Expression of Vascular Endothelial Growth Factor and Interleukin-6 in bile duct healing with autologous parietal peritoneum: a non-inferiority experimental study in rabbits

Nugroho Anung Noto notonugroho@student.uns.ac.id 1
Soetrisno Soetrisno 2
Mudigdo Ambar 3
Yarso Kristanto Yuli 4
Indarto Dono 5
Wahyudi Akmal Zhahir 6
1 Doctoral Program of Medical Sciences, Sebelas Maret University , Surakarta , Central Java , Indonesia
2 Department of Obstetrics and Gynecology, Dr. Moewardi Hospital/Faculty of Medicine, Sebelas Maret University , Surakarta , Central Java , Indonesia
3 Department of Anatomical Pathology, Dr. Moewardi Hospital/Faculty of Medicine, Sebelas Maret University , Surakarta , Central Java , Indonesia
4 Oncology Division, Surgery Department, Sebelas Maret University , Surakarta , Central Java , Indonesia
5 Department of Physiology and Biomedical Laboratory, Sebelas Maret University , Surakarta , Central Java , Indonesia
6 Faculty of Medicine, Sebelas Maret University , Surakarta , Central Java , Indonesia
Uversky Vladimir
Electronic publication date: 2025 Jun 9
Publication date: 2025
Volume: 13
Electronic Location ID: e19306
Received 2024 Oct 22; Accepted 2025 Mar 20
Copyright: ©2025 Nugroho et al.
Copyright year: 2025
Copyright holder: Nugroho et al.
License: This is an open access article distributed under the terms of the Creative Commons Attribution License, which permits unrestricted use, distribution, reproduction and adaptation in any medium and for any purpose provided that it is properly attributed. For attribution, the original author(s), title, publication source (PeerJ) and either DOI or URL of the article must be cited.
License URL: https://creativecommons.org/licenses/by/4.0/

Keywords: Bile duct injury, Laparoscopic cholecystectomy, Parietal peritoneum, IL-6, VEGF, Fibroblast

Funding: The authors received no funding for this work.

==============================
Background

Bile duct injury (BDI) remains a serious complication of hepatobiliary surgery, particularly in laparoscopic cholecystectomy, often leading to strictures, fibrosis, and long-term morbidity. Although traditional repair techniques such as primary suturing and Roux-en-Y hepaticojejunostomy are widely used, they carry risks of anastomotic complications and bile duct dysfunction. Autologous peritoneum has emerged as a potential alternative graft material due to its biocompatibility and regenerative properties. This study evaluates the efficacy of autologous parietal peritoneum graft in bile duct healing, focusing on fibroblast activity, vascular endothelial growth factor (VEGF), and interleukin-6 (IL-6) expression as indicators of tissue remodeling.

Materials and Methods

This experimental study involved 27 male New Zealand rabbits (Oryctolagus cuniculus), divided into three groups: group A (primary bile duct closure), group B (gallbladder transplant), and group C (parietal peritoneum transplant). The rabbits were anesthetized using a combination of ketamine, xylazine, isoflurane, and sevoflurane. Postoperative care included antibiotics and pain management. The study employed a post-test-only design. IL-6 and VEGF expression were assessed using enzyme-linked immunosorbent assay (ELISA), and the anastomosis was pathologically evaluated using hematoxylin and eosin (H&E) staining. Statistical analysis was conducted with SPSS 28.0, using one-way analysis of variance (ANOVA) or Kruskal–Wallis tests, with significance set at p < 0.05. Ethical approval was obtained.

Results

On day 3, fibroblast infiltration was significantly lower in the autologous peritoneum group (p = 0.040) compared to other groups, suggesting delayed initial recruitment. By day 7, fibroblast density increased, and by day 14, all groups exhibited well-organized tissue structures with no significant intergroup differences. VEGF (p = 0.788 on day 3, 0.473 on day 7, and 0.586 on day 14) and IL-6 (p = 0.629 on day 3, 0.587 on day 7, and 0.925 on day 14) levels showed no significant variations among the groups, indicating comparable angiogenic and inflammatory responses across treatment conditions.

Conclusion

Autologous peritoneum supports gradual bile duct healing, despite initial delayed fibroblast recruitment, with histological evidence of progressive tissue remodeling. However, the lack of significant differences in VEGF and IL-6 levels suggests that angiogenesis and inflammation were similarly regulated across groups. Given the study’s short follow-up period, further research is needed to assess the long-term integration, functional outcomes, and potential benefits of autologous peritoneum in bile duct reconstruction.

Introduction

The increasing utilization of laparoscopic cholecystectomy as the preferred treatment for gallbladder diseases has led to a rise in bile duct injury (BDI), a challenging complication in hepatobiliary surgery. The incidence of BDI varies by surgical approach, with laparoscopic cholecystectomy (LC) carrying a higher risk (0.4–0.6%) compared to open cholecystectomy (OC) (0.1–0.2%). Despite advancements in laparoscopic techniques and improved surgical experience, BDI still occurs in 0.2–1.3% of all cholecystectomies (Zidan et al., 2024). These injuries often result in significant morbidity due to impaired wound healing, leading to strictures, fibrosis, and prolonged recovery times (Kapoor, 2009).

Traditional bile duct repair methods, such as primary suture repair and hepaticojejunostomy, are associated with high risk of stricture rates and compromised bile duct vascularity, which necessitate novel surgical approaches to improve outcomes (Zielsdorf et al., 2019). The use of autologous tissues for surgical repair has gained attention due to their biocompatibility and lower rejection rates. Several autologous grafts frequently used in BDI management include the falciform ligament, round ligament of the liver, omental patch, inferior mesenteric vein, and vein grafts (Biglari, Van Den Bussche & Vanlangenhove, 2013; Ince et al., 2018; Dokmak et al., 2017; Ng & Kow, 2017; Xie et al., 2018). Alternative graft-based approaches, including vascularized autologous tissue grafts, decellularized matrices, and synthetic scaffolds, have been investigated but remain experimental and not widely adopted due to concerns over graft integration, long-term durability, and immune response (Nugroho et al., 2025). Among these, autologous parietal peritoneum graft has shown promise due to its rich vascularity and rapid integration with host tissues, yet its role in bile duct repair remains underexplored (Bonvini et al., 2012).

Wound healing in bile duct injuries is driven by angiogenic and inflammatory processes, where vascular endothelial growth factor (VEGF) and interleukin-6 (IL-6) play crucial roles in endothelial cell migration, proliferation, and immune response modulation. Fibroblasts also play a key role in bile duct healing by interacting with biliary epithelial cells (BECs) to regulate tissue repair. Following injury, fibroblasts become activated, proliferate, and contribute to extracellular matrix deposition, facilitating wound remodeling. However, excessive fibroblast activity can lead to fibrosis and ductal stenosis, which may compromise long-term bile duct function. Evaluating fibroblast density, alongside VEGF and IL-6 expression, provides deeper insight into the balance between effective tissue healing and the risk of fibrosis.

This study investigates the efficacy of autologous parietal peritoneum for bile duct repair in a New Zealand rabbit (Oryctolagus cuniculus) model, evaluating wound healing through the expression of VEGF, IL-6, and fibroblast density. Rabbits are well-established preclinical models due to their genomic and immune similarities to humans, making them valuable for translational research. By assessing the biological response of autologous parietal peritoneum in bile duct healing, this study aims to determine its potential to enhance surgical outcomes, reduce anastomotic leakage, and provide a foundation for future clinical applications in BDI management.

Materials & Methods

Animal model

An animal study was conducted with approval from the Institutional Review Board of the Ethics Committee of the Dr. Moewardi General Hospital, Surakarta, Indonesia with the number of 1.785/X/HREC/2023. The study included adult male New Zealand rabbits (Oryctolagus caniculus), aged 8–10 months and weighing between 2.0–3.0 kg. The New Zealand rabbits used in this study were obtained from the Prof. Soeparwi Veterinary Hospital, Faculty of Veterinary Medicine, Gadjah Mada University, an accredited breeding facility that adheres to ethical guidelines for animal care and use. Rabbits that were not in good condition were excluded based on predefined criteria. The minimum sample size was determined using Federer’s Formula, with at least nine rabbits per group. 41 rabbits that initially met our inclusion criteria underwent surgical procedures. However, 11 rabbits (26.82%), consisting of three from the control group, three from the parietal peritoneum group, and five from the vesica fellea group, were excluded from the study due to death before anastomosis tissue collection. The causes of death varied, but autopsy findings indicated that bile duct leakage was the primary cause in most cases.

Intervention groups

Rabbits were randomly divided into three groups of nine rabbits each. Group A (n = 9) act as a control and had the common bile duct primarily closed. Group B (n = 9) underwent cholecystectomy (1.5 cm length), insertion of a five cm tube as a stent, and transplantation of the parietal peritoneum. Group C (n = 9) had a cholecystectomy (1.5 cm length), insertion of a five cm tube as a stent, and transplantation of the gallbladder.

Intervention procedure

1. Pre-operative procedure. Following the Johns Hopkins University’s Animal Care and Use Committee (JHU ACUC) standards, rabbits were allowed to adapt to their new environment for 2–3 days before the surgical procedure. Rabbits were housed in individual cages with a minimum area of 0.5 square meters per animal, in a temperature-controlled environment (27 °C), with a 12-hour light/dark cycle. They were fed a standard rabbit diet consisting of pelleted food and provided with fresh water ad libitum. Blood samples were taken prior to surgery to assess blood chemistry. For pre-operative preparation, the rabbits were given albendazole at a dose of 20–50 mg/kg body weight (BW) orally to prevent worms and were fasted for 4–8 h before surgery. To further minimize confounding variables, all rabbits were kept in a controlled environment at a constant temperature of 27 °C. Additionally, their diets were standardized, and cage sizes were uniform to ensure that external factors did not influence the outcomes of the study.

2. Anesthetic procedure. According to Johns Hopkins University Animal Care and Use Committee (JHU ACUC) standards, rabbits were anesthetized with intravenous ketamine (50 mg/kg) and dexilasane (1–2 mg/kg), along with a 0.9% NaCl infusion. Ventilation was maintained with 2% halothane inhalation and 100% oxygen. The rabbits were positioned in a supine decubitus position, with anesthesia maintained using isoflurane and sevoflurane through a vaporizer.

3. Biliary-digestive-repair. After anesthetization, the rabbits were shaved, subjected to antisepsis, and placed in a sterile field. A 5-cm midline abdominal incision was made from the xiphoid process to the peritoneal cavity to expose and identify the common bile duct (CBD). A 15 mm × 5 mm excision was performed on the anterior wall of the CBD to create a defect. A similar excision was made on the parietal peritoneum harvested from the posterior abdominal wall to repair the defect (Figs. 1A, 1B). The graft was positioned over the CBD, and a 5-cm plastic stent (3.5 Fr NGT) was carefully inserted into the lumen to maintain patency (Fig. 1C). The biliodigestive anastomosis was secured using a 7-0 polypropylene simple interrupted suture, ensuring proper graft integration. The final anastomotic site showed a well-positioned graft with no immediate complications upon completion of the procedure (Fig. 1D).

4. Postoperative care. Postoperatively, the rabbits were given standard food and had free access to water, along with a 0.9% NaCl infusion. Intravenous atipamezole (five mg/kg) and xylazine (five mg/kg) in a 1:1 ratio were administered. The rabbits were kept in clean cages at a room temperature of 27 °C.

5. Animal termination. Rabbits were humanely euthanized on days 3, 7, and 14 post-treatment using an intravenous injection of phenobarbital at a dose of 100 mg/kg, following institutional ethical guidelines and the approved protocol. No rabbits required early euthanasia or met any criteria for premature termination. In cases where a rabbit died prematurely, it was excluded from the study. Biliary anastomosis tissue was taken to be examined by enzyme-linked immunosorbent assay (ELISA) for levels of IL-6 and VEGF. To ensure unbiased results, the laboratory analysts performing the ELISA, as well as all laboratory personnel involved, were blinded to the group assignments and specimen control. This blinding was strictly maintained throughout the analysis to prevent any potential bias in the evaluation of wound healing markers across the different treatment groups.

Immunoassay

The ELISA examination used the sandwich enzyme-linked immunosorbent assay (ELISA) method to quantify IL-6 and VEGF levels in tissue samples. The Rabbit IL-6 ELISA kit (MBS2507990, MyBioSource, Indonesia) and Rabbit VEGF ELISA kit (MBS765626, MyBioSource, Indonesia) followed the manufacturer’s protocol.

Figure 1 Stepwise surgical procedure in the autologous parietal peritoneum graft group.

A stepwise representation of the bile duct repair procedure using autologous parietal peritoneum grafts. (A) The peritoneal graft is excised from the posterior abdominal wall. (B) The harvested graft measures 15 mm × 5 mm. (C) A five cm plastic stent (3.5 Fr NGT) is inserted into the common bile duct lumen to maintain patency. (D) Final anastomotic appearance, showing a well-positioned graft with no immediate complications.

Sample preparation: tissue samples were cut, weighed, and homogenized in phosphate-buffered saline (PBS) (pH 7.4) at 4 °C. The homogenate was centrifuged at 14,000 rpm for 20 min, and the supernatant was collected for analysis.

Assay plates were prepared by adding 100 µL of standards or samples to designated wells, including a 0 pg/mL standard diluent as a negative control. The wells were sealed and incubated at 37 °C for 90 min. After incubation, biotinylated detection antibodies (100 µL per well) were introduced and incubated at 37 °C for 60 min. Following three washing cycles, 90 µL of horseradish peroxidase (HRP)-conjugate solution was added per well and incubated for 30 min at 37 °C. The plate was washed five times, and 100 µL of 3,3′,5,5′-Tetramethylbenzidine (TMB) substrate solution was added to initiate a colorimetric reaction. After 20 min of incubation at 37 °C in the dark, 100 µL of stop solution was added, changing the color from blue to yellow.

Optical density (OD) values were measured at 450 nm using an ELISA reader, and IL-6 and VEGF concentrations were determined based on standard curves.

This study employed a true experimental post-test-only control group design to investigate the effect of autologous parietal peritoneum grafts on biliary tract injury repair. Experiments were conducted between February and April 2023.

Fibroblast

The assessment of fibroblast activity was carried out through histological examination, utilizing hematoxylin and eosin (H&E) staining to observe tissue remodeling and healing processes in bile duct repair sites. To preserve both cellular and extracellular structures, tissue specimens were initially fixed in 10% buffered formalin for 24–48 h. Following fixation, the samples underwent a graded ethanol dehydration process (70%, 80%, 95%, and 100%), followed by xylene clearing to eliminate residual lipids and impurities. The specimens were then embedded in paraffin wax, creating stable blocks for further sectioning.

For microscopic evaluation, thin sections (five µm) were obtained from the paraffin-embedded samples using a microtome and then mounted onto clean glass slides. The slides were subjected to deparaffinization in xylene, after which they were progressively rehydrated through a series of descending ethanol concentrations (100%, 95%, 80%, and 70%). A final rinse in distilled water restored their hydrophilic properties. Hematoxylin staining was employed to highlight cell nuclei in blue, while eosin staining provided contrast by coloring the cytoplasm and extracellular matrix in shades of pink.

Microscopic observations were conducted under 400 × magnification using a light microscope. To quantify fibroblast density, fibroblasts were counted in five randomly selected high-power fields (HPFs) per sample, and the average number per HPF was determined as an indicator of tissue remodeling and wound healing. To ensure reliability, all slides were examined independently by two pathologists, and any discrepancies in fibroblast identification were resolved through collaborative evaluation and consensus discussions. Additionally, a qualitative histopathological analysis was performed to further assess tissue responses at the repair site.

Histopathology

The surviving rabbits underwent an exploratory laparotomy, during which the entire extrahepatic biliary system, including both grafts and native bile ducts, was excised. The collected bile duct specimens were promptly fixed in 10% neutral buffered formalin to preserve tissue integrity. The specimens were sent to the Pathology Department at Gadjah Mada University for histopathological evaluation. Hematoxylin and eosin (H&E) staining was conducted to assess the bile ducts’ morphological structures, providing detailed insight into tissue architecture and potential pathological changes.

Statistical analysis

All statistical analyses were conducted using SPSS version 28.0 (IBM Corp., New York, USA). The Shapiro–Wilk test was used to assess data normality, while Levene’s test evaluated variance homogeneity.

For normally distributed data with equal variances, one-way analysis of variance (ANOVA) was performed, followed by Bonferroni post-hoc analysis for pairwise comparisons. If data did not follow a normal distribution, the Kruskal–Wallis test was applied, with Dunn’s post-hoc test for multiple comparisons. When normality was met but variance homogeneity was violated, the Brown-Forsythe ANOVA was used, followed by Dunnett T3 post-hoc analysis to adjust for unequal variance.

Although sample size determination was based on Federer’s Formula, ensuring a minimum of nine rabbits per group, the study was primarily powered to detect early differences in IL-6, VEGF expression, and fibroblast activity. The statistical analysis included tests for normality and variance homogeneity, with non-parametric methods applied when necessary to maintain statistical robustness and ensure valid comparisons.

Parametric data were expressed as mean ± standard deviation (SD), while non-parametric data were presented as median and interquartile range (IQR). Statistical significance was determined using two-tailed tests, with a p-value <0.05 considered significant. Where applicable, effect sizes were calculated to provide a clearer understanding of the differences between groups.

Results

A total of 27 New Zealand rabbits were randomly allocated into three groups (n = 9 per group). Wound healing was evaluated by measuring IL-6 and VEGF expression at three postoperative time points (D-3, D-7, and D-14), alongside fibroblast counts to assess tissue regeneration. All surgical procedures were successfully completed without complications in the biliary-digestive anastomosis, and all rabbits survived the study duration.

IL-6 expression in Group 1 was consistently lower than in Group 2 and Group 3 across all time points; however, the differences were not statistically significant (p = 0.629 on D-3, p = 0.587 on D-7, and p = 0.925 on D-14) as summarized in Table 1 and Fig. 2. Similarly, VEGF expression did not show statistical significance between the groups (p = 0.788 on D-3, p = 0.473 on D-7, and p = 0.586 on D-14), as illustrated in Table 2 and Fig. 3.

Table 1 Analysis of IL-6 levels in different experimental groups over time.

Ductal IL-6 levels (pg/mL) are presented as mean ± standard deviation (SD) across three time points. Normality was confirmed using the Shapiro-Wilk test, and one-way ANOVA was used for statistical analysis. No significant differences in IL-6 levels were observed between groups (p > 0.05 at all time points) (n = 9 per group).

Time points	IL-6 levels (pg/mL), mean ± SD	p- value	
	Control group (n = 9)	Parietal peritoneum graft group (n = 9)	Gallbladder graft group C (n = 9)		
Day 3	138.27 ± 53.82	244.70 ± 166.83	190.64 ± 141.49	0.629a	
Day 7	111.15 ± 55.73	163.01 ± 8.96	276.84 ± 328.28	0.587a	
Day 14	83.65 ± 71.60	133.66 ± 150.02	101.05 ± 65.15	0.925a	
p-value	0.577a	0.578a	0.616a		
Notes.

a Analyzed using one-way ANOVA.

Figure 2 Comparison of mean IL-6 levels in different experimental groups over time.

Ductal IL-6 levels were measured on days 3, 7, and 14 in the autologous peritoneum graft, vesica fellea graft, and control groups. Data distribution was confirmed normal using the Shapiro–Wilk test, allowing for analysis using one-way ANOVA. No significant differences in IL-6 levels were observed across groups at any time point (p = 0.629 on day 3, p = 0.587 on day 7, and p = 0.925 on day 14). The autologous peritoneum group exhibited a non-significant trend toward higher IL-6 levels, suggesting an early localized inflammatory response. Data are presented as mean ± standard deviation (SD) (n = 9 per group).

Table 2 Analysis of VEGF levels in different experimental groups over time.

Ductal VEGF levels (pg/mL) are presented as mean ± standard deviation (SD) across three time points. Oneway ANOVA was performed, and the Brown–Forsythe ANOVA was used when variance homogeneity was violated. No significant differences were found between groups at any time point (p > 0.05) (n = 9 per group).

Time points	VEGF levels (pg/mL), mean ± SD	p- value	
	Control group (n = 9)	Parietal peritoneum graft group (n = 9)	Gallbladder graft group C (n = 9)		
Day 3	12.17 ± 8.97	13.05 ± 7.77	9.22 ± 2.16	0.788a	
Day 7	7.09 ± 1.48	8.56 ± 1.54	7.22 ± 1.55	0.473a	
Day 14	7.75 ± 3.24	6.58 ± 0.96	5.97 ± 1.02	0.586a	
p-value	0.557b	0.287a	0.126a		
Notes.

a Analyzed using one-way ANOVA test.

b Analyzed using Brown–Forsythe test.

Figure 3 Comparison of mean VEGF levels in different experimental groups over time.

Ductal VEGF levels were assessed on days 3, 7, and 14 in the autologous peritoneum, vesica fellea, and control groups. Normality was confirmed using the Shapiro–Wilk test. One-way ANOVA was applied for comparisons, and when variance homogeneity was violated, the Brown-Forsythe ANOVA was used. No significant differences were found in VEGF levels across groups (p = 0.788 on day 3, p = 0.473 on day 7, p = 0.586 on day 14). A non-significant trend of increased VEGF in the autologous peritoneum group suggests inosculation-driven vascularization. Data are presented as mean ± standard deviation (SD) (n = 9 per group).

For fibroblast counts, significant differences were observed on day 3 (p = 0.040), with Group 1 showing a mean count of 21.00 ± 5.51, compared to significantly lower counts in Group 2 (8.67 ± 4.98) and higher counts in Group 3 (23.67 ± 6.67). However, by day 7 and day 14, no significant differences were observed between the groups (p = 0.661 and p = 0.479, respectively), as shown in Table 3 and Fig. 4.

Table 3 Fibroblast count in different experimental groups over time.

Time points	Fibroblast count (fibroblasts/HPF at 400x), mean ± SD	p- value	
	Control group (n = 9)	Parietal peritoneum graft group (n = 9)	Gallbladder graft group C (n = 9)		
Day 3	21.00 ± 5.511	8.67 ± 4.982	23.67 ± 6.671	0.040a*	
Day 7	28.78 ± 7.41	21.56 ± 8.76	24.78 ± 28.78	0.661b	
Day 14	22.11 ± 4.07	23.33 ± 4.67	26.66 ± 5.77	0.479a	
p-value	0.430a	0.060a	0.731b		
Notes.

a Analyzed using ANOVA test.

b Analyzed using Kruskal–Wallis test.

1,2 Different superscript numbers indicate significant differences based on the LSD post hoc test (p < 0.05).

* Statistically significant difference (p < 0.05).

Figure 4 Comparison of fibroblast count in different experimental groups over time.

Fibroblast counts were evaluated at bile duct repair sites on days 3, 7, and 14 for the autologous peritoneum, vesica fellea, and control groups. Data are presented as mean ± standard deviation (SD). One-way ANOVA was used for parametric data, while the Kruskal–Wallis test was applied for non-parametric data. A significant difference was observed on day 3 (p = 0.040), prompting post-hoc least significant difference (LSD) analysis, which revealed that the autologous peritoneum group had significantly lower fibroblast counts than the control and vesica fellea groups (p < 0.05). No significant differences were found on days 7 and 14 (p > 0.05) (n = 9 per group).

These findings suggest that while there were observable differences in IL-6, VEGF, and fibroblast activity—particularly the significant fibroblast count on day 3—none of the groups demonstrated a statistically significant advantage in overall wound healing. This indicates that the use of autologous parietal peritoneum graft is not inferior to other interventions in promoting bile duct healing.

Histopathological analysis on day 3 showed significantly lower fibroblast infiltration in the autologous peritoneum group (p = 0.040) compared to other groups, suggesting delayed recruitment. Sparse fibroblasts were observed at this stage (Fig. 5A). By day 7, fibroblast density increased, indicating active granulation tissue formation (Fig. 5B). By day 14, fibroblast density and tissue organization had improved across all groups, with no significant intergroup differences. Histological changes showed clear evidence of tissue maturation (Fig. 5C). These findings indicate that autologous peritoneum grafts support progressive healing, despite an initial delay in fibroblast recruitment.

Figure 5 Representative histopathological analysis of bile duct repair tissue in the autologous peritoneum graft group.

Hematoxylin and eosin (H&E)-stained histopathological images at 300× magnification demonstrate tissue remodeling following bile duct repair. (A) Day 3: Sparse fibroblast infiltration, indicating the early healing phase. (B) Day 7: Increased fibroblast density, reflecting active granulation tissue formation. (C) Day 14: Well-organized tissue architecture, suggesting maturation and stabilization of fibroblast density.

Discussion

The incidence of BDI has increased alongside the widespread adoption of laparoscopic cholecystectomy (LC), which, despite advancements in technique, continues to pose a significant risk. BDI occurs in approximately 0.2–1.3% of all cholecystectomies, with laparoscopic procedures carrying a higher incidence (0.3–0.7%) compared to open surgery (Zidan et al., 2024). In the United States alone, 750,000 laparoscopic cholecystectomies are performed annually, underscoring the clinical burden of this complication (Pesce et al., 2019). The management of BDI depends on the severity and classification of the injury. Bile leakage (Strasberg type A) is often treated with primary bile duct suturing and endoscopic retrograde cholangiopancreatography (ERCP) stent placement (Chun, 2014; Garcés-Albir et al., 2019), whereas more complex injuries (Strasberg types E1–E3) typically require biliary reconstruction via Roux-en-Y hepaticojejunostomy (Garcés-Albir et al., 2019; Pitt et al., 2013). Although this procedure has high success rates and low stricture formation, potential complications include intestinal reflux, hormonal imbalances, and recurrent biliary infections (Xie et al., 2018; Cheung, Lorenzo Pisarello & LaRusso, 2018).

The autologous parietal peritoneum, derived from stem cells and capable of regeneration and transdifferentiation from mesothelial cells, offers regenerative and anti-inflammatory properties that are advantageous for bile duct repair. By promoting angiogenesis and reducing inflammatory responses, this approach highlights the complex biological processes involved in healing and underscores the potential for targeting these pathways in clinical interventions for bile duct injury (Bretón-Romero & Lamas, 2014; Dunnill et al., 2017).

In this study, we selected the parietal peritoneum as a graft because its surface mesothelium shares the same origin as the peritoneum and muscle layer of the duodenum, it is easily accessible without the need for additional incisions, and it is available in large quantities (Castillo et al., 2019). Partial bile duct injury triggers a critical sequence of cellular events, leading to cell fragmentation and activation of the body’s innate immune response, essential for tissue repair. In our study, the use of autologous parietal peritoneum graft provided a framework for cell healing and regeneration (Cheung, Lorenzo Pisarello & LaRusso, 2018). Reactive oxygen species (ROS) and transcription factor NF-κB play pivotal roles in wound healing by stimulating angiogenesis, cytokine production, and cell mobilization (Demidova-Rice, Hamblin & Herman, 2012; Sanchez et al., 2018).

A key limitation of this study is its short follow-up period, which only captures the early phase of wound healing and does not allow for evaluation of long-term graft integration, bile duct function, or potential late complications such as fibrosis or strictures. Future studies should extend the observation period to assess the durability of the repair, inflammatory response progression, and potential late adverse effects.

VEGF

A significant increase in VEGF signals angiogenesis, enhancing wound tissue perfusion and increasing oxygen availability (PO2), which supports cell proliferation and tissue repair (Shaik et al., 2020; Setiawan et al., 2022). In bile duct repair, VEGF plays a specialized role in promoting biliary epithelial cell (BEC) proliferation and modulating VEGFR-2 signaling, which facilitates interactions between inflammatory and mesenchymal cells, contributing to ductal healing (Mariotti et al., 2021).

Persistent inflammation can impair angiogenesis by increasing reactive oxygen species (ROS), activating NF-κB, suppressing VEGF expression, and inducing apoptosis (caspase-3) (Pereira Beserra et al., 2019; Mussbacher et al., 2023). In our study, ductal VEGF levels on days 3, 7, and 14 did not differ significantly between groups (p > 0.05), suggesting that all repair strategies provided comparable angiogenic support (Kim & Byzova, 2014). However, VEGF levels were higher in the autologous peritoneum group, likely due to inosculation-driven vascularization, where angiogenesis facilitates graft integration.

While VEGF is essential for biliary epithelial regeneration, excessive neovascularization can promote fibrosis and bile duct stricture formation, as observed in chronic biliary diseases (Mariotti et al., 2021). This highlights the delicate balance between sufficient angiogenesis for healing and excessive vascular remodeling, which may increase the risk of stenosis. Future studies should evaluate the long-term effects of VEGF-driven vascular remodeling in biliary anastomotic healing and fibrosis prevention (Mussbacher et al., 2023; Kim & Byzova, 2014; Johnson et al., 2020).

IL-6

IL-6 is a key regulator of inflammation and tissue remodeling, orchestrating immune cell recruitment, angiogenesis, and extracellular matrix modulation (Johnson et al., 2020). In bile duct repair, IL-6 plays a dual role: at controlled levels, it promotes biliary epithelial cell (BEC) proliferation and ductal remodeling, but prolonged IL-6 activity can lead to fibrosis and bile duct stricture formation (Bester & Pretorius, 2016).

IL-6 exerts its effects through JAK/STAT3 and NF-κB pathways, which regulate biliary epithelial survival, collagen deposition, and inflammatory resolution (Li et al., 2022). The presence of IL-6 and its receptor in human BECs suggests that IL-6 may exert both autocrine and paracrine effects, directly influencing biliary epithelium regeneration and fibrotic progression (Demetris et al., 2006).

In our study, ductal IL-6 levels on days 3, 7, and 14 did not show significant differences between groups (p > 0.05), suggesting that all repair strategies maintained a controlled inflammatory response. However, IL-6 levels appeared higher in the autologous peritoneum group, possibly reflecting a localized early inflammatory response that may have facilitated immune cell infiltration and wound remodeling. While this response could be beneficial for early epithelial repair, excessive IL-6-driven inflammation has been implicated in biliary fibrosis and chronic inflammation (Li et al., 2022).

This highlights the need for precise regulation of IL-6 activity—sufficient to drive bile duct healing but controlled to prevent fibrosis. Future studies should focus on IL-6 modulation strategies to optimize early inflammatory responses while mitigating long-term fibrotic progression, ensuring effective bile duct repair without pathological scarring.

IL-6 is a crucial regulator in wound healing, managing inflammatory responses, promoting tissue regeneration, aiding angiogenesis, and ensuring proper tissue remodeling. Its ability to balance pro-inflammatory and anti-inflammatory signals is vital for efficient and effective wound repair (Johnson et al., 2020). IL-6 signaling affects vascular integrity through endothelial activation, increased vascular permeability, immune cell recruitment, and vascular wall fibrosis. These effects reduce nitric oxide (NO) availability, as IL-6 influences endothelial nitric oxide synthase activity and expression while increasing superoxide production, which deactivates NO, affecting coagulation and platelet profiles (Bester & Pretorius, 2016; Didion, 2017). In the study, ductal IL-6 levels on days 3, 7, and 14 did not show significant differences between the treatment groups (p > 0.05), suggesting that excessive inflammation was absent in any groups. Although IL-6 levels appeared higher in the autologous parietal peritoneum group compared to the control, this trend did not reach statistical significance. This may indicate a potential early inflammatory response, which could facilitate immune cell recruitment and tissue repair, but further investigation is needed to confirm this hypothesis. However, while IL-6 plays a beneficial role in early healing, prolonged and excessive IL-6 expression could lead to chronic inflammation and fibrosis, highlighting the need for balanced inflammatory responses in effective healing (Johnson et al., 2020; Soliman & Barreda, 2023).

Fibroblast

The significantly lower number of fibroblasts in the autologous parietal peritoneum group on day 3 (p = 0.019) likely reflects an initial delay in fibroblast recruitment. This delay may be attributed to reduced fibrinolytic activity following peritoneal trauma, which impairs the conversion of plasminogen to plasmin, delaying fibrin degradation and slowing early wound healing processes. Additionally, delayed mesothelial cell regeneration—which plays a key role in initiating fibrinolytic activity from the wound bed—further contributes to this delay (Bakkum et al., 1996). Neutrophils produce Tumor Necrosis Factor (TNF)-α and Interleukin (IL)-1, which play a role in recruiting epithelial cells and fibroblasts to the wound area. The increase in fibroblast numbers occurs due to stimulation from fibroblastic growth factors (FGF), which are secreted as proteins during the platelet degranulation process (Goldberg & Diegelmann, 2010). By day 7, the number of fibroblasts increased in all groups, consistent with the proliferative phase of wound healing, indicating active tissue repair. Although the differences between groups were not significant, the steady increase in the autologous parietal peritoneum group suggests progressive integration and healing. During the proliferation phase, connective tissue and granulation tissue are formed due to fibroblast activation. Along with this, reepithelialization, neovascularization, and immunomodulation occur (Rodrigues et al., 2019). On day 14, the decrease in fibroblast numbers in the control group may indicate the onset of the remodeling phase, while the continuously higher numbers in the autologous parietal peritoneum and vesica fellea groups suggest ongoing tissue repair.

Histological analysis

The histological findings provide key insights into the healing dynamics of autologous peritoneum grafts in bile duct repair. The significantly lower fibroblast infiltration observed on day 3 suggests delayed initial recruitment, likely due to reduced fibrinolytic activity and slower mesothelial regeneration following peritoneal trauma. However, by day 7, fibroblast density increased, indicating a transition to the proliferative phase, characterized by granulation tissue formation and extracellular matrix remodeling.

By day 14, fibroblast distribution stabilized across all groups, with no significant intergroup differences, suggesting that the initial delay in fibroblast infiltration in the autologous peritoneum group did not impair long-term tissue remodeling. The presence of well-organized tissue architecture at this stage indicates that the graft successfully integrated and facilitated progressive healing. These findings suggest that autologous peritoneum grafts provide a controlled regenerative environment, promoting gradual tissue remodeling while potentially minimizing excessive fibroblast proliferation that could contribute to fibrosis and ductal stenosis. Thus, despite a slightly delayed early response, autologous peritoneum appears to support gradual and effective bile duct healing, reinforcing its potential as a viable graft material for biliary reconstruction.

Conclusions

This study evaluated the potential of autologous peritoneum as a graft for bile duct repair, assessing VEGF, IL-6 expression, and fibroblast density. The results indicate gradual but effective tissue remodeling, with initially delayed fibroblast recruitment followed by progressive healing. Although no significant differences were observed in VEGF and IL-6 levels, trends suggest that angiogenic and inflammatory responses were supported across all groups, reinforcing the graft’s ability to integrate into the bile duct environment.

However, the study is limited by its short follow-up period, which prevents conclusions about long-term graft function and integration. Further research is needed to assess its impact on biliary patency, fibrosis prevention, and anastomotic stability.

Despite these limitations, the findings provide preliminary evidence that autologous peritoneum could serve as a biocompatible alternative for bile duct reconstruction, warranting further investigation in long-term and functional studies.

Supplemental Information

Supplemental Information 1 Raw Data

The expression of VEGF, IL-6, and fibroblast in different treatment groups. This data is useful for further statistical analysis.

Supplemental Information 2 The ARRIVE Guidelines 2.0

The authors utilized ChatGPT to enhance the readability and clarity of the manuscript during its preparation. Following the use of this tool, the authors thoroughly reviewed and edited the content to ensure accuracy and take full responsibility for the final version of the publication.

Additional Information and Declarations

Competing Interests

Author Contributions

Animal Ethics

Data Availability

The authors declare there are no competing interests.

Anung Noto Nugroho conceived and designed the experiments, performed the experiments, analyzed the data, prepared figures and/or tables, authored or reviewed drafts of the article, and approved the final draft.

Soetrisno Soetrisno conceived and designed the experiments, performed the experiments, analyzed the data, prepared figures and/or tables, authored or reviewed drafts of the article, and approved the final draft.

Ambar Mudigdo conceived and designed the experiments, performed the experiments, analyzed the data, prepared figures and/or tables, authored or reviewed drafts of the article, and approved the final draft.

Kristanto Yuli Yarso conceived and designed the experiments, performed the experiments, analyzed the data, prepared figures and/or tables, authored or reviewed drafts of the article, and approved the final draft.

Dono Indarto conceived and designed the experiments, performed the experiments, analyzed the data, prepared figures and/or tables, authored or reviewed drafts of the article, and approved the final draft.

Akmal Zhahir Wahyudi conceived and designed the experiments, performed the experiments, analyzed the data, prepared figures and/or tables, authored or reviewed drafts of the article, and approved the final draft.

The following information was supplied relating to ethical approvals (i.e., approving body and any reference numbers):

The Ethics Committee of the Dr. Moewardi General Hospital, Surakarta, Indonesia approved this study (1.785/X/HREC/2023).

The following information was supplied regarding data availability:

The raw measurements are available in the Supplementary File.

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
