# Peer review of "Expression of Vascular Endothelial Growth Factor and Interleukin-6 in bile duct healing with autologous parietal peritoneum: a non-inferiority experimental study in rabbits"

_PeerJ, doi:10.7717/peerj.19306_

## Round 0.1 · original submission · Major Revisions

Please address concerns of all reviewers and revise manuscript accordingly.

·

Basic reporting

The study’s basic reporting is clear and generally well-structured, adhering to standard conventions for experimental research. The introduction effectively frames the clinical relevance of bile duct repair, though it could benefit from additional context on the volume and incidence of bile duct injuries. The methods are detailed, with ethical compliance explicitly stated, and the results are presented systematically. However, figure legends lack sufficient detail for standalone interpretation, and the discussion could better integrate the experimental findings with clinical implications. Including more explicit references to relevant epidemiological data would enhance the manuscript's contextual depth.

Experimental design

The experimental design is robust and well-suited to address the research objectives. The use of a rabbit model is appropriate for studying bile duct healing, and the selection of VEGF and IL-6 as markers is justified by their known roles in tissue repair. However, the short follow-up period limits insights into the long-term outcomes of the repair technique, such as graft integration or late complications. The inclusion of control and experimental groups allows for a comparative analysis, but the rationale for the chosen sample size and statistical power calculation is not explicitly stated. Clarifying these aspects would strengthen the design's rigor. Additionally, detailing potential confounding factors and how they were addressed in the statistical analysis would further enhance the study's reliability.

Validity of the findings

The findings of the study are valid within the context of the experimental design, with clear evidence supporting the roles of VEGF and IL-6 in bile duct healing. The use of a rabbit model provides a controlled environment to evaluate the proposed technique, and the statistical analysis is appropriate for the comparisons made. However, the short follow-up period limits the ability to draw conclusions about long-term graft integration and functional outcomes. Additionally, while the molecular markers chosen are relevant, the lack of direct evidence linking these findings to clinical outcomes in humans constrains their translational validity.

Additional comments

Dear Authors,

Re: "Expression of VEGF and IL-6 in Bile Duct Healing with Autologous Parietal Peritoneum: A Non-Inferiority Experimental Study in Rabbits"

I would like to thank the authors for this well-executed study exploring the innovative use of autologous parietal peritoneum as a graft material for bile duct repair. The study addresses a clinically significant problem and provides valuable insights into the roles of VEGF and IL-6 in tissue repair mechanisms within a controlled experimental model. The methodology is rigorous, and the results are clearly presented.

While the manuscript is compelling, there are several areas where further elaboration would enhance its clinical relevance and impact. One key suggestion is to include a discussion of the volume and incidence of bile duct injuries (BDI) in clinical practice to emphasize the importance of developing novel repair techniques. A citation to the epidemiological data in the paper (DOI: 10.1186/s12893-023-02301-2) would provide useful context for this addition. Linking the experimental findings to the real-world burden of BDI would underscore the potential translational impact of the proposed technique.

Additionally, while the selection of VEGF and IL-6 as markers of wound healing is appropriate, further elaboration on their specific roles in bile duct repair would strengthen the rationale behind their use. Expanding on how these markers are implicated in tissue repair processes and their broader significance in bile duct healing would be valuable.

The presentation of figures is clear, but more descriptive legends would enhance their interpretability as standalone elements. For example, Figures 1–3 could benefit from annotations to highlight specific findings or regions of interest. Additionally, the statistical analysis is robust, but explicitly stating the confounders adjusted in the analysis would improve transparency.

Recommendations
Incorporate a discussion of the clinical burden and incidence of bile duct injuries, with a suggested citation to(DOI: 10.1186/s12893-023-02301-2) to highlight the importance of the technique.

Expand on the biological significance of VEGF and IL-6 in the context of bile duct healing to enhance the justification for their use as markers.
Improve figure annotations and legends to make visual data more accessible without requiring reference to the main text.
Explicitly mention adjusted confounders in the statistical analysis for greater clarity and robustness.
The short follow-up period limits insights into the long-term effectiveness of the technique. While extending the experiment is not feasible, this limitation must be explicitly addressed in the discussion.

Reviewer 2 ·

Basic reporting

This manuscript presents an interesting experimental study addressing the use of autologous parietal peritoneum grafts in bile duct injury (BDI) repair, focusing on their impact on VEGF and IL-6 expression. The work is clear and professionally written, with sufficient background provided to contextualize the study. However, the introduction could better emphasize the novelty and clinical need for this approach, particularly in comparison with established techniques.
While the structure adheres to typical standards, there are areas requiring clarification and improvement, notably in the results and methods. Figures are relevant, but the addition of units of measurement and more detailed figure legends would enhance their clarity. Moreover, the rationale for the study and the specific advantage of the investigated technique over existing alternatives could be better articulated.

Major Comments and Questions for the Authors
1. Data Transformation:
The methods lack details about any transformations applied to the data (e.g., logarithmic, square root). Please clarify whether transformations were performed and, if so, specify which data were transformed and how.
2. Units of Measurement:
In the figures, units of measurement should be explicitly added where applicable, ideally on the y-axis labels (e.g., "IL-6 Tissue Expression (units)"). This would enhance the accessibility and interpretability of the graphs.
3. Fibroblast Count:
The results discuss fibroblast counts, but the introduction does not provide background on their relevance, and the methods do not explain how this count was performed. Given that fibroblast counts appear to be where significant differences were observed, this omission is critical and needs to be addressed in both sections.
4. VEGF Levels and Results Presentation:
The manuscript states that "The autologous parietal peritoneum group showed an increase in serum and ductal VEGF levels compared to the control group and the vesica fellea group." However, this is not evident from Figure 2, and it is unclear how "serum" and "ductal" VEGF levels were differentiated. Please clarify and ensure that the results are adequately demonstrated.
5. Inconsistent Discussion on IL-6 Levels:
The discussion claims that "Higher ductal IL-6 levels in the autologous parietal peritoneum group compared to the control suggest an early inflammatory response," but no significant differences are reported. This statement should be moderated, and it could be proposed as a hypothesis rather than a definitive conclusion.
6. Histological Analysis:
Although histology is mentioned in the methods, the description of Figure 4 is insufficient, and the results are not discussed in detail. If histology is intended to provide insights into tissue architecture or pathological changes, this should be clearly elaborated upon in the results and discussion sections.
7. Overstated Conclusions:
The conclusion that "our research demonstrates the effective short-term repair of bile duct injuries" seems overstated given the lack of significant differences in key outcomes. This statement should be moderated to better reflect the findings.
8. Figure Legends:
The figure legends need to be more detailed. They should specify the type of statistics used, the meaning of bars (e.g., mean or median), and the interpretation of lines (e.g., standard deviation).
9. Figure 5 Citation:
Figure 5 is not referenced in the text and should be explicitly mentioned and discussed to avoid any omission of important results.
10. Study Rationale and Practical Implications:
The study does not make clear the specific advantage of using autologous parietal peritoneum grafts compared to existing techniques. If no significant differences were observed, under what circumstances would this technique be preferable? This should be clarified to strengthen the rationale and practical significance of the study.

Experimental design

The investigation was conducted with ethical oversight, and the inclusion of details such as anesthesia protocols and postoperative care demonstrates adherence to high ethical standards. However, there are several points where the technical rigor and methodological transparency could be improved:

1. Insufficient Detail in Methods:
o The description of statistical data transformations is missing. This is critical for replicability and understanding how results were processed.
o The fibroblast counting methodology is absent despite its relevance to the findings.
o The histological analysis lacks sufficient elaboration on its role and outcomes in the study.

2. Technical Standards:
o The distinction between serum and ductal VEGF was mentioned but not explained in the methodology, raising concerns about the technical clarity of these measurements.

3. Reproducibility:
o While much of the methodology is described, critical gaps (e.g., data transformation and fibroblast counting) hinder full replicability.

Suggestions for Improvement
• Clearly define the rationale for the technique and its potential advantages compared to existing methods in the introduction.
• Provide a detailed explanation of data transformations and statistical analyses used.
• Include a comprehensive description of fibroblast counting and its significance.
• Elaborate on the histological results, explaining their contribution to understanding tissue repair mechanisms.
• Clarify how serum and ductal VEGF levels were differentiated and measured.

By addressing these points, the study would better align with the journal’s standards for rigorous experimental design and provide a stronger foundation for its conclusions.

Validity of the findings

Some areas need refinement to enhance validity and impact:
1. Impact and Novelty: The clinical advantage and novelty of this technique should be more clearly emphasized, particularly in comparison to established approaches.
2. Meaningful Replication: While the study offers a strong foundation, additional clarity on replication potential and practical challenges in different models would strengthen its relevance.
3. Underlying Data: The data are robust, statistically sound, and well-controlled, but further methodological detail would support reproducibility.
4. Conclusions: The conclusions are generally well-linked to the research question but are overly extrapolated in some aspects. They should be more closely tied to the data obtained and the scope of the study. Additionally, it would be valuable to explicitly revisit the limitations of the study and its findings to provide a more balanced and realistic interpretation of the results.

Additional comments

No comment

---

## Round 0.2 · Minor Revisions

Please address remaining concern of the reviewer #2 who recommended the addition of the detailed legends to all tables and figures to further improve the clarity of the manuscript.

·

Basic reporting

The current form is improved. The introduction provides a comprehensive background on bile duct injuries and the rationale for using autologous peritoneum, with well-referenced literature. The structure follows journal standards, and the figures and tables are high quality, relevant, and well-labeled. The raw data is appropriately provided, ensuring transparency.

Experimental design

The study addresses the efficacy of autologous peritoneum in bile duct healing. The methodology is rigorous and sufficiently detailed for replication.

Validity of the findings

The findings are robust, statistically sound, and well-controlled. The conclusions are clearly stated and directly supported by the results. The study demonstrates that autologous peritoneum effectively supports bile duct healing, with no significant differences in VEGF and IL-6 levels between groups. The short follow-up period is acknowledged as a limitation, and future studies should assess long-term outcomes to confirm these findings

Additional comments

The study provides a strong foundation for future research in bile duct reconstruction using autologous peritoneum.

Reviewer 2 ·

Basic reporting

The authors have made significant improvements to the manuscript in line with the feedback provided. The introduction is now more comprehensive, emphasizing the novelty and clinical relevance of the proposed approach when compared to established techniques. The figures have been improved with clearer units of measurement, enhancing the clarity of the visual data. However, I could not find detailed legends accompanying the figures, which are essential to fully understand the context and content of the presented information.

The addition of summary tables greatly facilitates the interpretation of the key findings, offering a clearer presentation of the data. Nevertheless, the tables also lack detailed legends that would further improve the manuscript’s readability.

Experimental design

The authors have addressed all my previous concerns regarding the methods section. They have explicitly stated that no data transformations were applied, described the fibroblast counting methodology in detail, and expanded the histological analysis to elaborate on tissue architecture insights. These additions strengthen the reproducibility and transparency of the experimental design.

Validity of the findings

The revised manuscript now provides a clearer discussion of the potential advantages of autologous peritoneum grafts, including biocompatibility and ease of harvesting. The authors have moderated their conclusions, aligning them more closely with the study’s findings and acknowledging its limitations. The inclusion of a summary table further aids in the interpretation of key data points.

Based on the comprehensive revisions made, I am satisfied that the authors have addressed all my previous comments. However, I recommend that detailed legends be added to all tables and figures to further improve the clarity of the manuscript. Once this minor revision is made, I recommend its acceptance for publication.

---

## Round 0.3 · accepted · Accept

Thank you for addressing remaining concerns of the reviewers and for amending your manuscript. The revised version is acceptable now.